# Comment on Zhong et al. Sugar-Sweetened and Diet Beverage Consumption in Philadelphia One Year after the Beverage Tax. *Int. J. Environ. Res. Public Health* 2020, *17*, 1336

**DOI:** 10.3390/ijerph182010926

**Published:** 2021-10-18

**Authors:** Wesley de Souza do Vale, Izabel Monteiro Dhyppolito, Silvana Chagas, Rosely Sichieri

**Affiliations:** Department of Epidemiology, Institute of Social Medicine, State University of Rio de Janeiro, Rio de Janeiro 20550-013, Brazil; izabelmdh@gmail.com (I.M.D.); silvanachagasestistica@gmail.com (S.C.); rosely.sichieri@gmail.com (R.S.)

In February 2020, the paper “Sugar-Sweetened and Diet Beverage Consumption in Philadelphia One Year after the Beverage Tax” was published in the International Journal of Environmental Research and Public Health. This quasi-experimental study investigated the impact of beverage taxes on consumption in Philadelphia in a population-based sample [1]. The study compared the consumption of diet and sweetened beverages with Philadelphia residents’ sugar with neighboring cities and similar cities that do not border Philadelphia. The main finding of the study was that the Philadelphia beverage tax had no major impact on residents’ consumption of sugar-sweetened diet drinks and bottled water after one year of implementation.

We agree that the object of investigation of this study is extremely relevant, and it is its strong point. A recent systematic review by the Cochrane Group, points with moderate evidence that the increase in the price of such beverages is associated with a decrease in sales [2]. Critical reviews also suggest that taxes on sugary drinks can be effective [3], so the subject of this study is of great relevance. However, some methodological errors were made and need to be highlighted.

The major problem of the study is the loss of follow-up, which is associated with the lack of a paragraph on the sample calculation method. According to the checklist presented by CONSORT 2010 [4], among the 25 items listed is the need to report on the methods used to determine the sample size. However, as seen in the article, it is not possible to see the criteria adopted by the researchers to select a sample (n = 2767) of the population of interest to the study, with analysis based on 515 participants.

Furthermore, it is of utmost importance that experimental or quasi-experimental study protocols are registered, as they ensure that they fulfill a minimum set of recommendations in the course of their development [4]. Therefore, the International Committee of Medical Journal Editors (ICMJE) strongly recommends that authors register their respective studies before initiating [5]. However, we do not see the record of this article.

As the results of this study were negative, the small sample size has no power to conclude that there is no association between taxation and change in beverage consumption. Additionally, a recent systematic review by the Cochrane Group, although not including studies with taxation [the subject of a separate review protocol not yet published] [6], points out with moderate evidence that the increase in the price of such drinks is associated with a decrease in sales [2]. Critical reviews also suggest that taxes on sugary drinks can be effective [3]. It is important to state that results found seem to diverge from what is reported in the literature, without power to test it. Given the importance of these findings for public policy, it is our opinion that the results of the present study with such low power should be interpreted with caution.

## Data Availability

Not applicable.

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
