# Peer review of "Comment on Zhong et al. Sugar-Sweetened and Diet Beverage Consumption in Philadelphia One Year after the Beverage Tax. Int. J. Environ. Res. Public Health 2020, 17, 1336"

_ijerph, 2021, doi:10.3390/ijerph182010926_

Round 1

Reviewer 1 Report

The letter to the editor presented for review concerns an article dealing with the important topic of the consumption of sweetened and dietetic beverages in connection with the introduction of the sugar tax.

The authors of the letter rightly point out some shortcomings in the description of the methodology of the experiment and recommend caution in drawing far-reaching conclusions.

The presented comments are debatable and can certainly help other authors in better planning and describing similar experiments.

Author Response

This reviewer recommends acceptance. We are grateful for the considerations presented.

Reviewer 2 Report

This letter focuses on a very important issue of a published article entitled “Sugar-Sweetened and Diet Beverage Consumption in Philadelphia One Year after the Beverage Tax”. I have carefully read the manuscript of the letter and published the article. My observations are as follows: -The concerns raised in the letter are valid. -Each point has a scientific background. -The sample size was really small, particularly for concluding remarks against a sensitive issue like taxation. -Though the published article has already mentioned these limitations in their limitation section [page 9 under discussion section], the researcher should be alert in the future to avoid these limitations for explaining sensitive issues. The overall arguments of the letter are okay and acceptable. It can help researchers to be cautious to explain the research findings in the future.

Author Response

(The authors gave the same response as above.)

Reviewer 3 Report

The comment can be improved. I suggest that the authors comment on both the strengths and weaknesses of the paper. Also, please ensure that each paragraph has at least three sentences, with the first sentence of each paragraph summarizing the paragraph.

Author Response

We consider that the object of investigation of the study in question is its strong point and we report this in the first paragraph, and that the loss to follow-up is its weak point, as discussed in the second paragraph. We are grateful for the considerations presented.